# *PRPH2*-Related Retinal Dystrophies: Mutational Spectrum in 103 Families from a Spanish Cohort

**DOI:** 10.3390/ijms25052913

**Published:** 2024-03-02

**Authors:** Lidia Fernández-Caballero, Inmaculada Martín-Merida, Fiona Blanco-Kelly, Almudena Avila-Fernandez, Ester Carreño, Patricia Fernandez-San Jose, Cristina Irigoyen, Belen Jimenez-Rolando, Fermina Lopez-Grondona, Ignacio Mahillo, María Pilar Martin-Gutierrez, Pablo Minguez, Irene Perea-Romero, Marta Del Pozo-Valero, Rosa Riveiro-Alvarez, Cristina Rodilla, Lidya Rodriguez-Peña, Ana Isabel Sánchez-Barbero, Saoud T. Swafiri, María José Trujillo-Tiebas, Olga Zurita, Blanca García-Sandoval, Marta Corton, Carmen Ayuso

**Affiliations:** 1Department of Genetics & Genomics, Instituto de Investigación Sanitaria-Fundación Jiménez Díaz University Hospital, Universidad Autónoma de Madrid (IIS-FJD, UAM), 28040 Madrid, Spain; lidia.fernandezc@quironsalud.es (L.F.-C.); inmaculada.martinm@quironsalud.es (I.M.-M.); fblancok@quironsalud.es (F.B.-K.); aavila@quironsalud.es (A.A.-F.); fermina.lopez@quironsalud.es (F.L.-G.); pablo.minguez@quironsalud.es (P.M.); crodilla.ext@quironsalud.es (C.R.); ana.sbarbero@quironsalud.es (A.I.S.-B.); stahsin@quironsalud.es (S.T.S.); mjtrujillo@fjd.es (M.J.T.-T.); olga.zurita@fjd.es (O.Z.); 2Center for Biomedical Network Research on Rare Diseases (CIBERER), Instituto de Salud Carlos III, 28029 Madrid, Spain; 3Department of Ophthalmology, Fundación Jiménez Díaz University Hospital, 28040 Madrid, Spain; ester.carreno@quironsalud.es (E.C.); bjimenezr@salud.madrid.org (B.J.-R.); mariap.marting@quironsalud.es (M.P.M.-G.); bgarcia@fjd.es (B.G.-S.); 4Department of Genetics, Ramón y Cajal University Hospital, 28034 Madrid, Spain; 5Instituto Ramón y Cajal de Investigación Sanitaria (IRYCIS), 28034 Madrid, Spain; 6Ophthalmology Service, Donostia University Hospital, 20014 Donostia-San Sebastián, Spain; 7Department of Statistics, Instituto de Investigación Sanitaria-Fundación Jiménez Díaz University Hospital, Universidad Autónoma de Madrid (IIS-FJD, UAM), 28040 Madrid, Spain; imahillo@fjd.es; 8Bioinformatics Unit, Instituto de Investigación Sanitaria-Fundación Jiménez Díaz University Hospital, Universidad Autónoma de Madrid (IIS-FJD, UAM), 28040 Madrid, Spain; 9Sección de Genética Medica, Servicio de Pediatría, HCU Virgen de la Arrixaca, 30120 Murcia, Spain

**Keywords:** retinitis pigmentosa, cone/cone–rod dystrophy, macular dystrophy, *PRPH2*, genotype–phenotype correlation

## Abstract

*PRPH2*, one of the most frequently inherited retinal dystrophy (IRD)-causing genes, implies a high phenotypic variability. This study aims to analyze the *PRPH2* mutational spectrum in one of the largest cohorts worldwide, and to describe novel pathogenic variants and genotype–phenotype correlations. A study of 220 patients from 103 families recruited from a database of 5000 families. A molecular diagnosis was performed using classical molecular approaches and next-generation sequencing. Common haplotypes were ascertained by analyzing single-nucleotide polymorphisms. We identified 56 variants, including 11 novel variants. Most of them were missense variants (64%) and were located in the D2-loop protein domain (77%). The most frequently occurring variants were p.Gly167Ser, p.Gly208Asp and p.Pro221_Cys222del. Haplotype analysis revealed a shared region in families carrying p.Leu41Pro or p.Pro221_Cys222del. Patients with retinitis pigmentosa presented an earlier disease onset. We describe the largest cohort of IRD families associated with *PRPH2* from a single center. Most variants were located in the D2-loop domain, highlighting its importance in interacting with other proteins. Our work suggests a likely founder effect for the variants p.Leu41Pro and p.Pro221_Cys222del in our Spanish cohort. Phenotypes with a primary rod alteration presented more severe affectation. Finally, the high phenotypic variability in *PRPH2* hinders the possibility of drawing genotype–phenotype correlations.

## 1. Introduction

Inherited retinal dystrophies (IRDs) are a group of rare diseases with a prevalence of 1:3000–4000 individuals worldwide [1,2]. These conditions arise due to a genetic etiology, leading to dysfunction of photoreceptor and retinal pigment epithelium (RPE) cells, resulting in severe visual impairment or blindness. IRDs account for 5% of blindness in the Western world, being the major cause of vision loss in children and young adults [3].

There are more than 300 genes and loci associated with these pathologies (RetNet, https://web.sph.uth.edu/RetNet/, accessed on 29 December 2023). *PRPH2* (MIM *179605) is one of the most frequent non-syndromic IRD (NS-IRD) disease-causing genes, accounting for 3.4% of individuals with the disease in Japan [4], 3% in the USA and Canada [5] and 4.6% of families in the UK [6]. In Spain, pathogenic variants in *PRPH2* are found in approximately 4% of families [7]. Currently, 318 pathogenic variants in *PRPH2* have been described according to the Human Gene Mutation Database (HGMD^®^ 2023.4, https://digitalinsights.qiagen.com/products-overview/clinical-insights-portfolio/human-gene-mutation-database/, accessed on 29 December 2023).

*PRPH2*, also called retinal degeneration slow (*RDS*), is located on chromosome 6p21.1. It encodes for peripherin-2 (PRPH2), a member of the tetraspanin family of proteins, which contains four transmembrane domains, a cytoplasmatic loop (C-loop) and two intradiscal loops (D1-loop and D2-loop) [8]. PRPH2 is expressed primarily in the rim regions of rod and cone outer-segment discs and lamellae, being essential for their proper formation [9,10,11]. Pathogenic mutations in *PRPH2* can lead to a wide spectrum of IRD presentations, including retinitis pigmentosa (RP), cone/cone–rod dystrophy (CD/CRD), and macular dystrophy (MD), as previously reported [12,13]. *PRPH2* is associated with high phenotypic variability even in patients belonging to the same family [14,15,16].

The aim of this work is to analyze the *PRPH2*-associated mutational and clinical spectrum in one of the largest cohorts recruited worldwide, and to describe genotype–phenotype correlations and novel pathogenic variants.

## 2. Results

We collected genetic data on 220 patients carrying disease-causing variants of *PRPH2* from 103 unrelated families with IRD. In addition, we gathered ophthalmic histories from 129 individuals, fundus image description from 100 patients and/or the electrophysiological examination from 83 of the 220 patients.

### 2.1. Ophthalmic Characteristics of PRPH2 Patients

Table 1 summarizes the clinical features of 91 of the 220 individuals carrying pathogenic *PRPH2* variants for whom age at onset (AAO) of subjective symptoms data were available, and we performed their phenotypic classification into two clinical groups: non-RP and RP. A total of 38 of the 220 patients included in this study were referred to in our center as “IRD affected”, lacking specific phenotype information. Consequently, they were not included in the “RP” or “non-RP” groups. Out of 182 patients, there were 16 that were asymptomatic at the age of the last examination (16, 9%) and who were ascertained during family segregation after identifying a pathogenic *PRPH2* variant in a close relative. Despite the high phenotypic variability of our cohort, with an ample range of disease onset from the 1st to the 7th decade (Table 1), clinical data among symptomatic cases showed a lower number of patients with RP (35, 19%) compared to non-RP presentations (131, 72%).

### 2.2. Mutational Spectrum of PRPH2 Variants in Our Spanish Cohort

We identified 56 different variants in *PRPH2*, being heterozygous in 102 of the families, showing autosomal dominant inheritance, and homozygous in 1 family. Twelve variants were first reported by our team in previous studies [17,18,19,20]. The variants identified in our cohort included 34 missense, 11 frameshift indels, 7 nonsense, 2 in-frame indels, 1 splicing and 1 copy number variant (CNV) (Table 2). Missense variants were the most frequent in our cohort, accounting for 64% (67/104) of probands alleles (Figure 1A), followed by truncating variants (28%, 29/104).

The location of *PRPH2*-variants in the protein domains and associated phenotypes observed in our cohort are represented in Figure 1B. There was a clear enrichment of disease-causing *PRPH2* variants in the D2-loop domain, which harbors 77% of the disease-causing coding SNVs found in our cohort total (Figure 1C). As shown in Figure 1D, this domain was significantly enriched in non-truncating alleles (86%) versus truncating alleles (63%) (*p*-value = 0.0003). Specifically, there are several variants close to crucial cysteine residues involved in forming disulfide bounds between Cys165, Cys213, Cys214 and Cys222 (Figure 1B). Additionally, it is observed that there is an enrichment of the RP phenotype in this region, although it was not statistically significant.

In our cohort, we identified several recurrent variants (Table 2), accounting for 64% of the families. The most recurrent variant was the previously reported p.Gly167Ser that is present in seven unrelated families with autosomal dominant CD/CRD and/or MD. The next two most frequent variants were p.Gly208Asp and p.Pro221_Cys222del, and each were carried by six unrelated families. Interestingly, this latter variant had only been described to date in Spanish patients without any specific regional preference [19,50]. Additionally, the variant p.Leu41Pro which accounted for 4.8% of the total alleles, was identified in five unrelated families from the Basque Country, a region in northern Spain.

To analyze a possible founder effect for these two recurrent variants, p.Pro221_Cys222del and p.Leu41Pro, we performed haplotype analysis using 10 SNPs. Remarkably, a common haplotype for each variant was shared exclusively among the *PRPH2* mutation carriers (Appendix A). These two haplotypes were formed using a different combination of genotypes for these 10 SNPs and both spanned a minimum of 0.3 Mb (chr6: 42929839-42665888, hg37) between rs835 and rs7760250. Appendix A shows that r2 values fit the SNPs involved in these two common haplotypes, indicating that these loci are in linkage equilibrium and are not coinherited. These results indicated the presence of a shared haplotype for p.Pro221_Cys222del and for p.Leu41Pro, suggesting that these mutations have a very likely ancestral origin in the carried families.

#### Novel Disease-Causing Variants and *PRPH2*-VUS Identified in Our Cohort

In this work, we identified 11 novel variants that were considered to be associated with the phenotype according to ACMG guidelines, in silico predictions and, when available, family segregation (Appendix A).

First, we identified six novel variants, four of them in the D2-loop domain and two in the 4th transmembrane domain, which are expected to lead to a truncated protein. The novel frameshift variants c.562del, c.567dup, c.623del, c.649_650insTAGCTGCTGCAATCCTA, c.809_817delinsCCTTCGAGGTA and the novel nonsense p.Trp273* variant were not identified in population databases. Additionally, p.Trp273* was segregated in an affected relative.

In addition, we found five novel non-truncating variants that are predicted to cause a deleterious effect by destabilizing protein structure. All these variants had not previously been reported in population databases, and in silico predictors classified them as damaging (Appendix A). First, we identified two novel missense variants affecting p.Gly38, a highly conserved residue on the first transmembrane domain of PRPH2. The variant c.112G>A, which substitutes the uncharged Gly38 to the bigger polar-positive-charged Arginine (p.Gly38Arg) in the protein core, was found in a proband with RP, and her two affected siblings referred to our service with initial diagnoses of RP and MD, respectively. A second variant c.113G>A, which changes this non-polar Gly38 to a polar-negative-charged Glutamic acid (p.Gly38Glu), was carried by two unrelated probands referred with MD.

Additionally, we have found two unreported non-synonymous aspects variants in the hotspot D2-loop domain. The novel missense c.637T>G introduces a non-polar Glycine in place of the highly conserved Cys213, which is involved in forming a crucial disulfide bridge with Cys166 for keeping the proper protein structure. The variant c.637T>G was segregated in the affected son. The missense variant c.745G>C replaces the uncharged Gly249 with a polar Arginine that would lead to a steric clash in the protein core of PRPH2. This residue is also involved in the likely pathogenic variant Gly249Ser reported previously [51]. The novel variant Gly249Arg was carried by two affected relatives from the same family with MD. Finally, the in-frame variant c.633_656del, which leads to the loss of nine highly conserved residues p.(Phe211_Pro219delinsLeu) in the hot-spot D2-loop region, including Cys213 and Cys214, was not present in population databases.

Moreover, the variant c.904G>A (p.Glu302Lys) in the C-terminal domain was identified in two affected individuals of the family PRPH2-099. This variant was previously reported in ClinVar as a VUS and not found in gnomAD population databases. Although it changes a negatively charged Glutamic acid residue using a positively charged Lysine, computational prediction tools unanimously supported a benign effect of this variant. Both the proband and her daughter presented with RP, but it was not possible to perform a segregation analysis on other non-affected relatives. Finally, we have considered that this variant remains a VUS because there is insufficient evidence to reliably support its implication as the cause of RP in this family (Appendix A).

### 2.3. Genotype–Phenotype Correlations

Due to the high phenotypic variability observed in our cohort, we performed a genotype–phenotype analysis to identify possible correlations between the type (truncating/non-truncating) and protein location (inside/outside D2-loop) of the variant and the AAO of the first symptom. Figure 2A shows statistically significant differences between patients with RP and patients with non-RP in relation to the AAO (*p*-value < 0.005). Figure 2B shows a trend between the protein consequence related to disease onset, with patients with non-truncating SNVs presenting earlier onset (*p*-value < 0.1). Regarding protein location, there were no statistically significant differences, but a trend was observed for patients with variants located inside the D2-loop to have an earlier AAO (Figure 2C).

Phenotypic variability was observed for carriers of the same variant. As shown in Figure 1B, eight variants have been associated in our cohort with different diagnoses of IRD (RP and non-RP). In this sense, we evaluated the high phenotypic variability associated with the recurrent variant p.Gly208Asp in our cohort that was carried by 11 patients from six unrelated families who were referred for genetic testing with different diagnoses (Appendix A). Interestingly, for two unrelated patients (PRPH2-057 and PRPH2-058), the AAO of first symptoms was earlier than in five carriers with available clinical data, whose first symptoms were developed during their third–sixth decade.

First, the patient PRPH2-057 was a sporadic case born into an endogamic family with no relevant history of ocular disease that carried homozygously p.Gly208Asp. The disease onset in this patient was in the second decade. No ophthalmic data was available from relatives; therefore, incomplete penetrance cannot be excluded for this variant, which is similar to what has been previously reported [52] (Figure 3A).

The patient PRPH2-058-1 showed their first symptom in their first decade. As shown in Figure 3A, the patient at 56 years old showed typical retinal features of RP, with no macular involvement. In addition to the heterozygous *PRPH2* variant, IRD genetic testing identified two additional pathogenic variants *IMPG2*, c.513T>G; p.Tyr171* and c.2322G>A; p.Trp774 (NM_016247.4). The family-segregation analysis for *IMPG2* showed that both variants were compound heterozygous (Figure 3B). The proband’s mother, who also carried the variant p.Gly208Asp in *PRPH2*, was diagnosed with macular degeneration at 60 years old. The diagnosis of early-onset retinitis pigmentosa in this patient was even earlier than observed in the *PRPH2*-related patients with RP of our cohort (Table 1), which seems to be a compatible phenotype with *IMPG2* causative variants (Figure 4A).

The *PRPH2*-related different severity was also observed even in the same patient. As shown in Figure 4B, the patient PRPH2-018-5, who carried the variant p.Arg142Trp, presented with macular dystrophy with different severities between both eyes on funduscopy and OCT. This patient had suffered with night blindness and photophobia since her youth, and dyschromatopsia only in RE. The patient reported visual acuity loss (VAL) at 53 years old, with 1.7 in the RE and 0.8 in the LE (logMAR).

## 3. Discussion

In this work, we describe the largest cohort of unrelated IRD families carrying disease-causing variants in *PRPH2* recruited from a single center. According to Wang Y. et al. around 1000 families have been described worldwide with *PRPH2*-associated retinopathy [53]; therefore, the current Fundación Jiménez Díaz (FJD) cohort, composed of 103 families, accounts for about 10% of the total number of IRD-described families.

We identified 56 different variants in the FJD-cohort, of which 41% (23/56) were first identified in this cohort, 11 in this current work and 12 in previous studies [17,18,19,20]. Additionally, this work supports the pathogenicity for the variants p.Arg220Gly first reported in ClinVar as likely pathogenic, and c.708C>A;p.Tyr236* first reported in LOVD (VKGL data sharing initiative Nederland) as pathogenic, but they have not been reported in the literature previously. The variant p.Arg220Gly affects the same residue as other pathogenic variants previously described [43,54,55]. The variant c.708C>A has the same consequence (p.Tyr236*) as the previously described variant c.708C>G [4,36]. Currently, there are 318 *PRPH2* variants associated with IRD (Human Gene Mutation Database, HGMD^®^ 2023.4, last access December 2023), so we identified in our cohort almost 1/5 of the variants previously described. Furthermore, 19 of the 56 variants in our cohort are recurrent, i.e., identified in two or more families, accounting for 63% of allele frequency; thus, these variants are responsible for more than half of our Spanish cohort.

As in other reported studies [4,5,6], *PRPH2* is one of the most frequent genes in our current cohort of 5124 families with IRD, accounting for 4% of non-syndromic IRD families [7]. However, there are mutational differences between the different cohorts studied according to ethnicity (Table 3). In our Spanish cohort, the most frequent variants were p.Gly167Ser, p.Gly208Asp and p.Pro221_Cys222del, whereas in Chinese, Japanese and USA cohorts, the variants p.Gly305Alafs*19, p.Arg142Trp and c.828+3A>T, accounted for 33.3%, 13.3% and 17.4% of alleles, respectively [46,53,56]. By contrast, these frequent variants were a minority in our cohort at 1.9%, 3.9% and 0%, respectively. Similarly, the common variant p.Arg172Trp reported in several cohorts worldwide [39,43,46,53,56,57,58], only accounted for 1.9% of alleles in ours.

In this work, we identified some variants that appear to be unique to the Spanish population. First, a novel in-frame variant p.Pro221_Cys222del was recurrently present in our cohort in six unrelated families, accounting for 5.8% of total alleles. To our knowledge, this variant has been only reported in Spanish patients [19,50]. Additionally, the variant p.Leu41Pro was present in five unrelated families (4.8% allele frequency), all of them from the Basque Country. For both variants, haplotyping analysis indicated the presence of a shared haplotype in carriers associated with each pathogenic variant, suggesting that they arose from, very likely, a founder effect. Concerning the variant p.Leu41Pro and its identification in five unrelated families from the Basque Country, the geographical isolation that has historically occurred in this region of Northern Spain supports its likely founder effect. Based on the work of Irene Perea-Romero et al., 2021, it is known that 5% (285/6,089) of the families in our IRD-cohort are referred from the Basque Country (see Figure 1 in that article). Therefore, these data add further credibility to the results.

The prevalence of missense *PRPH2*-variants in our cohort (64% of the total) is close to those of previous reports, 47% to 63% [16,46]. Similarly to the reported data [16,46], most of the pathogenic variants, including the vast majority of missense, were located in the D2-loop protein domain (77%), which mediates the crucial interaction with ROM1 for proper formation of the photoreceptor outer segment [59,60]. Specifically, most variants are located near four cysteine residues, Cys165, Cys213, Cys214 and Cys222, crucial for forming disulfide bounds for proper folding and subunit assembly [61], highlighting their importance in protein function.

Our findings also show a significantly earlier AAO in patients with RP than in the subgroup of non-RP. Interestingly, it is also observed that there is an enrichment of RP phenotypes in the cysteine residues region, although this is not statistically significant, supporting the important role of this region. However, due to the fewer cases presenting with RP in the *PRPH2*-cohort (19%) and the presence of the non-RP phenotype also in this region, a larger sample size would be necessary. A study published by Ikelle et al. revealed that the toxicity of the mutant protein and reduced protein levels affects more rods than cones [62], which could be associated with an earlier onset in the RP-related patients from our cohort. However, our results show high phenotypic variability among patients, even those belonging to the same family or in the same individual, as reflected by the different severity between RE and LE. Consequently, the phenotypic variability associated with *PRPH2* variants makes it difficult to draw more extensive genotype–phenotype correlations.

Furthermore, we identified two different genotypes for the variant p.Gly208Asp that modulated disease onset in two patients from unrelated families. In our cohort, patients who only carry this pathogenic variant in the heterozygous state had an AAO of first symptoms between their third and sixth decade, whereas one homozygous patient from our cohort, with no family history, debuted symptoms in their second decade. This variant has recently been reported in a CRD family with incomplete penetrance [52]. Similarly, Wang et al. also reported two homozygous patients for the *PRPH2* variants Cys213Arg and p.Leu185Pro, with Leber congenital amaurosis or juvenile RP [63]. Our findings also support that biallelic *PRPH2* variants are responsible for more severe early-onset retinal dystrophy. Moreover, another patient in our cohort with the same variant p.Gly208Asp also had an earlier onset of the first symptoms in their first decade. However, this patient also carried biallelic pathogenic variants in *IMPG2*. Recessive variants in this gene are associated with the presentation of early-onset RP [64]. Further investigation is needed in order to clarify the effect of the *PRPH2* variant in this particular case. The identification of disease-causing variants in more than one IRD-associated gene poses an additional challenge to determine the exact pathogenic progression of the disease due to possible modifying effects with one another. For this reason, we believe more research in this direction will help to further elucidate these phenomena.

In conclusion, here, we describe the largest cohort of patients with *PRPH2*-associated IRD from a single center, in which we have found a very likely founder effect for two recurrent variants, and we describe genotype–phenotype correlations. These findings expand our understanding about the mutational and phenotypic spectrum of *PRPH2*, relevant not only to the Spanish population but also on a global scale, considering the extensive Spanish diaspora, especially in Latin America.

### Study Limitations

This *PRPH2*-cohort is part of our IRD-cohort, recruited by FJD [7], which exhibits a skewed recruitment pattern, with more individuals from Madrid and its surrounding areas (such as Castile and Leon, Castile-La Mancha, and Extremadura). This bias likely stems from our hospital and has been the reference center for these regions. Despite these limitations, our IRD-cohort, and, therefore, also our *PRPH2*-cohort, present a sizable sample size and a comprehensive molecular analysis performed over the years, making our study representative of the Spanish population.

Additionally, ophthalmic examinations of the patients were performed at different Spanish centers. Therefore, the classification of patients without examination data accessible in FJD was based on the reason for referral.

## 4. Materials and Methods

### 4.1. Subjects

A total of 220 patients from 103 unrelated families carrying disease-causing variants in *PRPH2* were included in our study. These families were identified from a cohort of 5124 unrelated families with IRD referred for genetic testing to the Genetics Service of the Fundación Jiménez Díaz University Hospital (FJD, Madrid, Spain) from different hospitals throughout Spain from 1990 to December 2023. Patients carrying disease-causing variants in *PRPH2* were selected from the local clinical database. Available genetic and ophthalmological data were obtained from a retrospective review of electronic medical records.

All subjects signed informed consent before participating. The study project was approved by the FJD Research Ethics Committee (Approval No.: PIC172-20_FJD, 16 September 2020) and fulfills the tenets of the Declaration of Helsinki and its further reviews.

### 4.2. Clinical Classification

For 129 of the 220 patients with molecular diagnoses, clinical data were collected from a self-reported ophthalmic history recorded from questionnaires. Ophthalmic exam data, such as electrophysiology and/or description of fundus images were also collected from 83 and 100 patients, respectively. A detailed examination of 57 patients was performed at the Department of Ophthalmology of the FJD, including best-corrected visual acuity, Humphrey visual field testing, electroretinography testing, and clinical imaging, including fundus color imaging, fundus autofluorescence, and spectral-domain optical coherence tomography.

Patients were classified using the available ophthalmic data and/or the “reason of referral” with functional or morphological criteria into two subgroups: (i) RP, which included patients with signs of peripheral rod dysfunction, and (ii) non-RP, which included patients with CD/CRD and/or MD. Several patients subjectively reported having no symptoms at the age they were examined, but those carrying pathogenic *PRPH2* variants previously identified in their affected relatives were referred to as asymptomatic (A) in Appendix A. Functional criteria were based on ERG recordings or, when unavailable, on reported clinical manifestations, and morphological criteria were based on the description of fundus images.

### 4.3. Molecular Diagnosis and Analysis of Variants

*PRPH2* variants were identified using different molecular approaches: (a) commercial genotyping arrays (Asper Biotech, Tartu, Estonia), (b) direct Sanger sequencing of coding regions of *PRPH2*, (c) multiplex ligation-dependent probe amplification (MLPA) kits for *PRPH2* (MRC-Holland, Amsterdam, the Netherlands), and (d) next-generation sequencing (NGS), including custom IRD gene panels, clinical exome sequencing (CES) and whole-exome sequencing (WES), as previously described [17,18,54]. In addition, screening for known family pathogenic variants and segregation studies, when possible, was performed using Sanger sequencing.

The variant interpretation was based on the American College of Medical Genetics and Genomics (ACMG) guidelines [65], in silico predictions and family segregation. The variants found in *PRPH2* were explored in different databases: (i) Human Gene Mutation Database, HGMD^®^ 2023.4, (https://digitalinsights.qiagen.com/products-overview/clinical-insights-portfolio/human-gene-mutation-database/, accessed on 29 December 2023), (ii) ClinVar (https://www.ncbi.nlm.nih.gov/clinvar/, accessed on 29 December 2023) and (iii) Leiden Open Variation Database v.3.0, LOVD, (https://www.lovd.nl/, accessed on 29 December 2023).

The potential effect of variants of uncertain significance (VUS) and novel missense variants was assessed in silico using functional pathogenicity, conservation, and protein stability predictors including the Ensembl Variant Effect Predictor [66] and DynaMut2 [67].

### 4.4. Haplotype Analysis

To identify common ancestral haplotypes, genotypes of 10 informative single-nucleotide polymorphisms (SNPs) markers (Appendix A) for a region 0.3 Mb surrounding *PRPH2* (rs835, rs434102, rs425876, rs3818086, rs6928781, rs7764439, rs3763236, rs9471969, rs10948059 and rs7760250) were ascertained using the available CES data from four patients carrying the variant p.Leu41Pro and four carrying the variant p.Pro221_Cys222del. Haplotypes were compared with four unrelated patients carrying other *PRPH2* pathogenic variants. The pairwise linkage disequilibrium (LD) data of these SNPs in the Iberian population were explored using the Ensembl Linkage Disequilibrium Calculator [68] (Appendix A).

### 4.5. Statistical Analysis

Phenotypic features considered for statistical analysis were the self-reported AAO of the first symptom, including visual acuity (VA) and visual field (VF) loss, metamorphopsia, photophobia and night blindness (NB). Categorical and continuous data were expressed as proportions and the median, respectively. Genotypes were stratified according to (a) variant type, considering truncating (frameshift indels, splicing, and nonsense changes) versus non-truncating variants (missense and in-frame indels), and (b) protein location, as inside versus outside the D2-loop domain. Statistical analysis was assessed using Kaplan–Meier analysis of ophthalmological symptoms event-free survival and compared using a log-rank test. Comparisons of the distribution of non-truncating and truncating alleles inside and outside the D2-loop found in probands were assessed by applying Fisher’s exact test. *p*-values < 0.05 were considered statistically significant. Statistical analyses were performed using R version 4.2.2, including the library of survival for survival analyses.

## Figures and Tables

**Figure 1 ijms-25-02913-f001:**
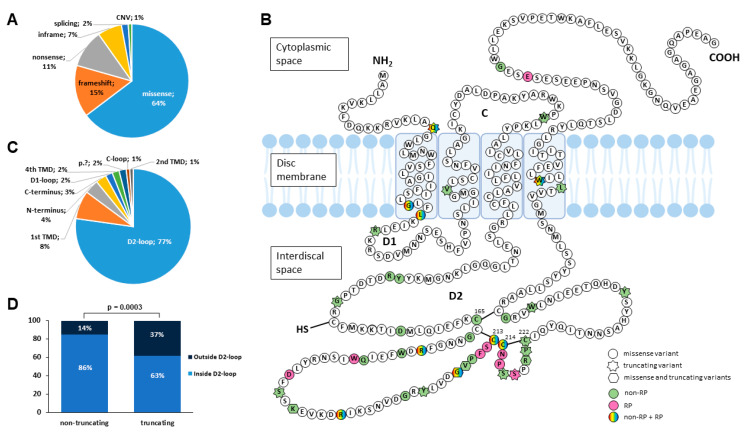
Variant distribution in the *PRPH2* Spanish cohort. (**A**) Distribution of alleles by type of variant, considering 103 probands. (**B**) Location of disease-causing coding variants in the protein structure and associated phenotypes. (**C**) Distribution of proband alleles by protein domain location. (**D**) Distribution of non-truncating and truncating alleles found in probands inside and outside the D2-loop domain (*p*-value = 0.0003). CNV: copy number variant; C: cytoplasmatic loop; D1: intradiscal loop D1 (D1-loop); D2: intradiscal loop D2 (D2-loop); RP: retinitis pigmentosa.

**Figure 2 ijms-25-02913-f002:**
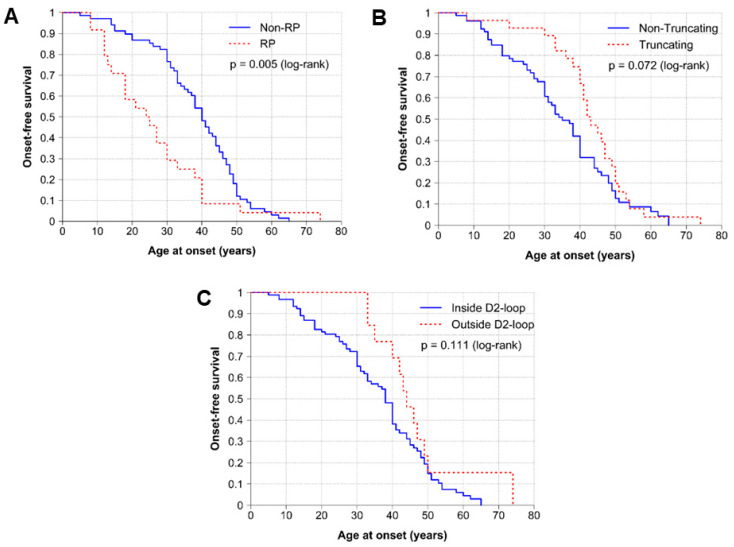
Genotype–phenotype correlations. Long-rank test and survival curve (onset-free) of the AAO of the first symptom of retinopathy, considering (**A**) the phenotypic presentation between retinitis pigmentosa (RP) and non-RP (MD and CR/CRD) subgroups; (**B**) the type of variant, truncating (frameshift, nonsense and splicing) versus non-truncating (missense, in-frame) variants; (**C**) the location of coding variants in the D2-loop domains: outside D2-loop versus inside D2-loop.

**Figure 3 ijms-25-02913-f003:**
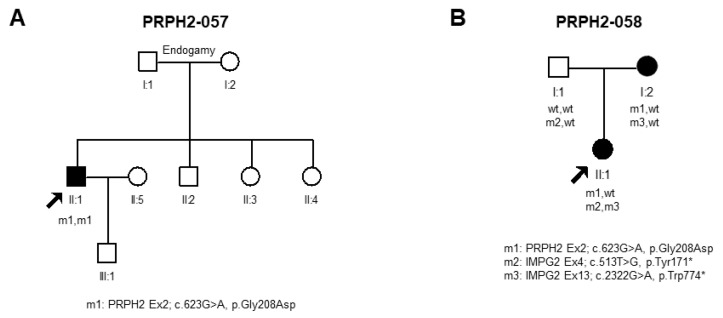
Pedigrees and segregation of families with atypical genotypes. (**A**) PRPH2-057, carrying homozygously p.Gly208Asp, and (**B**) PRPH2-058 carrying 3 different alleles in 2 genes, *PRPH2* and *IMPG2*. Arrows indicate the proband in each family. m: pathogenic alleles; wt: wild-type alleles.

**Figure 4 ijms-25-02913-f004:**
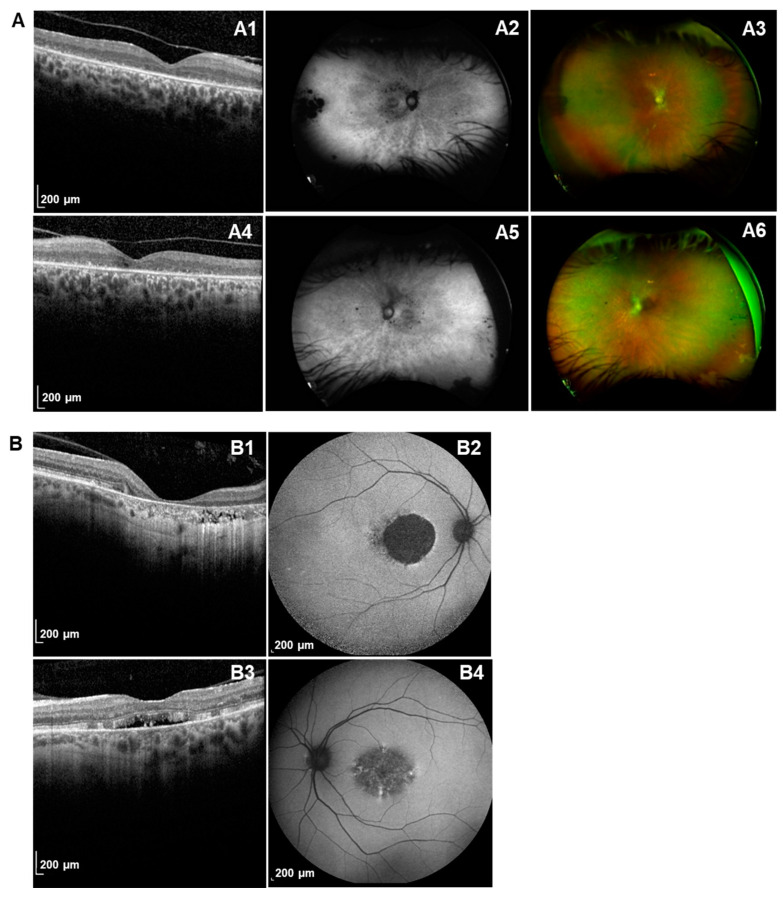
Retinal imaging of patients carrying *PRPH2*. Tomographic, fundoscopic and fundus autofluorescence (FAF) of right eye (RE) and left eye (LE) for 2 patients with disease-causing *PRPH2* variants. (**A**) Patient PRPH2-058 is a 56-year-old female, with a diagnosis of early-onset retinitis pigmentosa, carrying the prevalent variant p.Gly208Asp in *PRPH2*, and biallelic variants in *IMPG2* ((A1–A3): RE; (A4–A6): LE). OCT depicted the absence of extensive outer layers in both eyes affecting the fovea without cystic macular oedema. Fundoscopy revealed pale papilla with vascular attenuation, macula without pigmentary alterations, and absence of bone spicules but scattered pigmentary alterations in the periphery of both eyes. FAF images showed peripapillary atrophy with mottled hypoautofluorescence of the posterior pole, which was more evident in arcades, patches of atrophy in the extreme temporal periphery in RE and some small patches of atrophy in the extreme temporal periphery in LE. (**B**) Patient PRPH2-018-5 is a 61-year-old female, carrying the variant p.Arg142Trp, with a diagnosis of MD, for which clinical imaging shows a different severity in RE (B1–B2) and LE (B3–B4). OCT depicted well-delimited macular atrophy with the disappearance of outer retinal layers in RE and detachment of subfoveal neuroepithelium with thickening of the ellipsoid layer at this level in LE. FAF images showed well-demarcated hypoautofluorescence plaque affecting fovea with surrounding hyperaurofluorescence at its nasal rim and mottled hypo/hyperautofluorescence at its temporal rim in RE and central hypoautofluorescence with areas of reticular hyperautofluorescence within it in LE.

**Table 1 ijms-25-02913-t001:** Clinical characteristics of patients in our *PRPH2*-associated cohort. MD: macular dystrophy; CR/CRD: cone dystrophy/cone–rod dystrophy; RP: retinitis pigmentosa; A: asymptomatic; AAO: age at onset; VAL: visual acuity loss; VFL: visual field loss; NB: night blindness.

Patients’ Characteristics	Non-RP		RP	A
	MD	CD/CRD		
**TOTAL (%)**	**103 (57)**	**28 (15)**	**35 (19)**	**16 (9)**
Male (no. total)	43	14	11	6
Female (no. total)c	60	14	24	9
**AAO (no. patients with data/total)**	**51/103**	**16/28**	**24/35**	**-**
Median	40	41	24.5	-
Range	8–65	5–62	8–74	-
**VAL, no. patients with data**	**41**	**13**	**16**	**-**
Median	40	42	36.5	-
Range, years	8–65	5–62	12–74	-
**VFL, no. patients with data**	**13**	**10**	**19**	**-**
Median	45	46.5	35	-
Range, years.	25–60	5–65	12–74	-
**NB, no. patients with data**	**16**	**11**	**19**	**-**
Median	40.5	35	24	-
Range, years.	20–65	25–62	8–51	-

**Table 2 ijms-25-02913-t002:** *PRPH2* disease-causing variants identified in FJD-cohort. TMD: transmembrane domain; ACMG: American College of Medical Genetics; P: pathogenic; LP: likely pathogenic; VUS: variants of uncertain significance; Het.: heterozygous state; Homo.: homozygous state.

Exon	Nucleotide Change	Protein Change	Protein Domain	ACMG Classification	Type of Variant	Allele Count (%)	Het.	Homo.	Double Diagnosis	References
1	c.52C>T	p.Gln18*	N-terminus	LP (PVS1, PM2)	nonsense	4 (3.85)	4	0	0	Del Pozo-Valero, 2022 [20]
1	c.112G>A	p.Gly38Arg	1st TMD	LP (PM2, PP1, PP2, PP3)	missense	1 (0.96)	1	0	0	**This study**
1	c.113G>A	p.Gly38Glu	1st TMD	LP (PM2, PP2, PP3)	missense	2 (1.92)	2	0	0	**This study**
1	c.122T>C	p.Leu41Pro	1st TMD	LP (PM2, PP3, PP2)	missense	5 (4.81)	5	0	0	Peeters, 2021 [16]; Bianco, 2023 [21]
1	c.136C>T	p.Arg46*	D1-loop	P (PVS1, PM2, PP5)	nonsense	2 (1.92)	2	0	0	Meins, 1993 [22]
1	c.205del	p.Val69Cysfs*30	2nd TMD	P (PVS1, PM2, PP5)	frameshift	1 (0.96)	1	0	0	Manes, 2015 [23]
1	c.290G>A	p.Trp97*	C-loop	P (PVS1, PM2, PP5)	nonsense	1 (0.96)	1	0	0	Antonelli, 2022 [24]
1	c.421T>C	p.Tyr141His	D2-loop	P (PM1, PM2, PM5, PP2, PP3, PP5)	missense	1 (0.96)	1	0	0	Trujillo, 2001 [25]
1	c.424C>T	p.Arg142Trp	D2-loop	LP (PM1, PM2, PM5, PP2, PP5)	missense	4 (3.85)	4	0	0	Hoyng, 1996 [26]
1	c.441del	p.Gly148Alafs*5	D2-loop	P (PVS1, PM2, PP5)	frameshift	1 (0.96)	1	0	0	Trujillo, 1998 [27]
1	c.469G>A	p.Asp157Asn	D2-loop	P (PM1, PM2, PM5, PP2, PP3, PP5)	missense	1 (0.96)	1	0	0	Jacobson, 1996 [28]
1	c.493T>C	p.Cys165Arg	D2-loop	P (PM1, PM2, PM5, PP2, PP3, PP5)	missense	1 (0.96)	1	0	0	Simonelli, 2007 [29]
1	c.499G>A	p.Gly167Ser	D2-loop	P (PM1, PM2, PM5, PP2, PP3, PP5)	missense	7 (6.73)	7	0	0	Testa, 2005 [30]
1	c.514C>T	p.Arg172Trp	D2-loop	P (PM2, PM5, PP3, PP2, PP5)	missense	2 (1.92)	2	0	0	Wells, 1993 [31]
1	c.515G>A	p.Arg172Gln	D2-loop	LP (PM1, PM2, PM5, PP2, PP5)	missense	2 (1.92)	2	0	0	Wells, 1993 [31]
1	c.520T>A	p.Trp174Arg	D2-loop	P (PM1, PM2, PM5, PP2, PP3, PP5)	missense	1 (0.96)	1	0	0	Peeters, 2021 [16]
1	c.533A>G	p.Gln178Arg	D2-loop	LP (PM1, PM2, PP2, PP3, PP5)	missense	1 (0.96)	1	0	0	Sohocki, 2001 [32]
1	c.536G>T	p.Trp179Leu	D2-loop	P (PM1, PM2, PM5, PP2, PP3, PP5)	missense	1 (0.96)	1	0	0	Fernandez-San Jose, 2015 [18]
1	c.556G>A	p.Asp186Asn	D2-loop	LP (PM1, PM2, PM5, PP2, PP5)	missense	1 (0.96)	1	0	0	Kitiratschky, 2011 [33]
1	c.562del	p.Ser188Profs*68	D2-loop	LP (PVS1, PM2)	frameshift	4 (3.85)	4	0	0	**This study**
1	c.567dupC	p.Lys190Glnfs*28	D2-loop	LP (PVS1, PM2)	frameshift	1 (0.96)	1	0	0	**This study**
1	c.568A>G	p.Lys190Glu	D2-loop	LP (PM1, PM2, PP2)	missense	1 (0.96)	1	0	0	Falsini, 2022 [34]
IVS1	c.582-1G>A	p.?	-	P (PVS1, PM2, PP5)	splicing	2 (1.92)	2	0	0	Fernandez-San Jose, 2015 [18]
2	c.(581+1_582-1)_(828+1_829-1)del	p.?	D2-loop to 4th TMD	LP (PVS1, PM2)	CNV	1 (0.96)	1	0	0	Weisschuh, 2020 [35]
2	c.584G>A	p.Arg195Gln	D2-loop	P (PM1, PM2, PM5, PP2, PP3, PP5)	missense	1 (0.96)	1	0	0	Alapati, 2014 [36]
2	c.584G>T	p.Arg195Leu	D2-loop	P (PM1, PM2, PM5, PP1, PP2, PP5)	missense	4 (3.85)	4	0	0	Yanagihashi, 2003 [37]
2	c.605G>A	p.Gly202Glu	D2-loop	LP (PM1, PM2, PP2, PP5)	missense	1 (0.96)	1	0	0	Maggi, 2021 [38]
2	c.609_625del	p.Tyr204Profs*8	D2-loop	P (PVS1, PM2, PP5)	frameshift	1 (0.96)	1	0	0	Trujillo, 1998 [27]
2	c.623del	p.Gly208Alafs*48	D2-loop	LP (PVS1, PM2)	frameshift	1 (0.96)	1	0	0	**This study**
2	c.623G>A	p.Gly208Asp	D2-loop	P (PM1, PM2, PM5, PP2, PP3, PP5, PS4)	missense	7 (6.73)	4	2	1	Kohl, 1997 [39]
2	c.625G>A	p.Val209Ile	D2-loop	LP (PM1, PM2, PM5, PP2, PP5)	missense	1 (0.96)	1	0	0	Coco, 2010 [40]
2	c.626del	p.Val209Alafs*47	D2-loop	P (PVS1, PM2, PP5)	frameshift	1 (0.96)	1	0	0	Martin-Merida, 2019 [19]
2	c.628C>T	p.Pro210Ser	D2-loop	P (PM1, PM2, PM5, PP2, PP3, PP5)	missense	1 (0.96)	1	0	0	Kemp, 1994 [41]
2	c.631T>C	p.Phe211Leu	D2-loop	LP (PM1, PM2, PP2, PP3, PP5, PS1)	missense	1 (0.96)	1	0	0	Manes, 2015 [23]
2	c.633_656del	p.Phe211_Pro219delinsLeu	D2-loop	LP (PM1, PM2, PM4)	inframe	1 (0.96)	1	0	0	**This study**
2	c.634A>G	p.Ser212Gly	D2-loop	P (PM1, PM2, PM5, PP2, PP3, PP5, PS4)	missense	4 (3.85)	4	0	0	Farrar, 1992 [42]
2	c.637T>C	p.Cys213Arg	D2-loop	P (PM1, PM2, PM5, PP2, PP3, PP5)	missense	1 (0.96)	1	0	0	Payne, 1998 [43]
2	c.637T>G	p.Cys213Gly	D2-loop	LP (PM1, PM2, PM5, PP2, PP3)	missense	1 (0.96)	1	0	0	**This study**
2	c.638G>T	p.Cys213Phe	D2-loop	P (PM1, PM2, PM5, PP2, PP3, PP5)	missense	2 (1.92)	2	0	0	Villaverde, 2007 [44]
2	c.641G>A	p.Cys214Tyr	D2-loop	P (PM1, PM2, PM5, PP2, PP3, PP5)	missense	2 (1.92)	2	0	0	Trujillo, 2001 [25]
2	c.643A>T	p.Asn215Tyr	D2-loop	P (PM1, PM2, PM5, PP2, PP3, PP5)	missense	1 (0.96)	1	0	0	Martin-Merida, 2018 [17]
2	c.646C>T	p.Pro216Ser	D2-loop	P (PM1, PM2, PM5, PP2, PP3, PP5)	missense	1 (0.96)	1	0	0	Fishman, 1994 [45]
2	c.649_650insTAGCTGCTGCAATCCTA	p.Ser217Ilefs*45	D2-loop	LP (PVS1, PM2)	frameshift	3 (2.91)	3	0	0	**This study**
2	c.653C>A	p.Ser218*	D2-loop	P (PVS1, PM2, PP5)	nonsense	1 (0.96)	1	0	0	Reeves, 2020 [46]
2	c.658C>G	p.Arg220Gly	D2-loop	P (PM1, PM2, PM5, PP2, PP3, PP5)	missense	1 (0.96)	1	0	0	ClinVar
2	c.658C>T	p.Arg220Trp	D2-loop	P (PM1, PM2, PM5, PP2, PP3, PP5)	missense	3 (2.91)	3	0	0	Payne, 1998 [43]
2	c.658del	p.Arg220Glyfs36*	D2-loop	P (PVS1, PM2, PP5)	frameshift	1 (0.96)	1	0	0	Boon, 2007 [47]
2	c.660_665del	p.Pro221_Cys222del	D2-loop	LP (PM1, PM2, PM4, PP5)	inframe	6 (5.77)	6	0	0	Martin-Merida, 2019 [19]
2	c.708C>A	p.Tyr236*	D2-loop	LP (PVS1, PM2)	nonsense	1 (0.96)	1	0	0	LOVD
2	c.708C>G	p.Tyr236*	D2-loop	P (PVS1, PM2, PP5)	nonsense	1 (0.96)	1	0	0	Strom, 2012 [48]
2	c.734_737dupTGTG	p.Trp246Cysfs*56	D2-loop	LP (PVS1, PM2)	frameshift	1 (0.96)	1	0	0	Del Pozo-Valero, 2022 [20]
2	c.745G>C	p.Gly249Arg	D2-loop	LP (PM1, PM2, PM5, PP1, PP2, PP3)	missense	1 (0.96)	1	0	0	**This study**
2	c.809_817delinsCCTTCGAGGTA	p.Leu270Profs*9	4th TMD	LP (PVS1, PM2)	frameshift	1 (0.96)	1	0	0	**This study**
2	c.818G>A	p.Trp273*	4th TMD	LP (PVS1, PM2, PP1)	nonsense	1 (0.96)	1	0	0	**This study**
3	c.904G>A	p.Glu302Lys	C-terminus	VUS (PM2, PP2, BP4)	missense	1 (0.96)	1	0	0	ClinVar
3	c.914G>A	p.Gly305Asp	C-terminus	LP (PM2, PP2, PP5)	missense	2 (1.92)	2	0	0	Felbor, 1997 [49]

**Table 3 ijms-25-02913-t003:** Comparison of the frequency of recurrent variants in different cohorts from China, Japan and the USA. n: number of studied families.

Variant	Chinese Cohort (Wang, 2023) [53] (n = 15)	Japanese Cohort (Oishi, 2021) [56] (n = 30)	USA Cohort (Reeves, 2020) [46] (n = 161)	Spanish Cohort (This Study) (n = 103)
**c.122T>C (p.Leu41Pro)**	Not reported	Not reported	Not reported	**4.8%**
**c.424C>T (p.Arg142Trp)**	Not reported	13.3%	3.1%	**3.9%**
**c.499G>A (p.Gly167Ser)**	Not reported	6.7%	0.6%	**6.7%**
**c.514C>T (p.Arg172Trp)**	6.7%	13.3%	5.6%	**1.9%**
**c.599T>A (p.Val200Glu)**	Not reported	10.0%	Not reported	**Not found**
**c.623G>A (p.Gly208Asp)**	Not reported	Not reported	0.6%	**6.7%**
**c.660_665del (p.Pro221_Cys222del)**	Not reported	Not reported	Not reported	**5.8%**
**c.828+3A>T (p.?)**	Not reported	Not reported	17.4%	**Not found**
**c.914G>A (p.Gly305Alafs*19)**	33.3%	Not reported	Not reported	**1.9%**

## Data Availability

Genomic data are available on the European Genome-phenome Archive (EGA).

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
