# Peer review of "PRPH2-Related Retinal Dystrophies: Mutational Spectrum in 103 Families from a Spanish Cohort"

_ijms, 2024, doi:10.3390/ijms25052913_

Round 1

Reviewer 1 Report

Comments and Suggestions for Authors

Authors present a genetic study focusing on PRPH2-related retinal dystrophies and the mutational spectrum observed in 103 families from a Spanish cohort. The research is well-structured, featuring an appropriate introduction, methodology for data generation, results description, discussion, and conclusion. However, it is uncertain whether similar studies have not been conducted by other research groups. The identification of the variant p.Leu41Pro in 5 unrelated families (with a 4.8% allele frequency), all from the Basque Country, is particularly intriguing. However, the absence of a comprehensive list detailing the distribution of families across different Spanish regions is a notable limitation. Nevertheless, this finding has been corroborated by a published article (https://www.ncbi.nlm.nih.gov/pmc/articles/PMC7754416/), lending further credibility to the results. The authors appropriately acknowledge the need for additional research in this area to better understand these phenomena. Overall, the study is satisfactorily presented and could be considered acceptable for publication.

Reviewer 2 Report

Comments and Suggestions for Authors

Reviewer report for Fernandez-Caballero et al.  PRPH2-related retinal dystrophies: mutational spectrum in 103 2 families from a Spanish cohort

The authors report on a large group of patients with molecularly confirmed PRPH2-IRD in a Spanish registry. They provide comprehensive phenotypic and genotypic data including novel variants, founder effects in the Basque region and confirming phenotypic variability within the population. This is an important addition to the literature and understanding of PRPH2-IRD. The paper is well written and technically very sound showing the expertise of the authors.  There are a few minor comments and a few additions which could further strengthen the manuscript. Well done!

My comments are below:

Abstract

Lines 33/34: did patients with RP have a specific variant type.

Intro: very well written, concise and contains all relevant references.

Results:

Line 75: AAO – spell out at first use.  In this journal’s format, because results come before methods, the first abbreviation needs to occur here.

Line 77: should the % read 8%?  ie 16/200 = 0.08

Lines 81/82: recheck your %.  Is this out of a denominator of 200? If so, it should read 17.5% RP and 65.5% non-RP.  This only adds to 182…

Table 1. In the 2nd patient column, should this read CD/CRD instead of CR/CRD?

Line 87: Please clarify.  For the heterozygous cases, are these all AD inheritance or are there some compound heterozygous AR cases?

What was the ethnic origin of this PRPH2 cohort? Were they all Spanish?

Line 121: Interesting! The haplotype data is a great addition. Well done.

Line 136: Add ‘the’ in front of ‘D2-loop’ and in front of ‘4th transmembrane.’

Line 191: Suggest change to ‘carriers OF the same variant.’

Line 205: Suggest amending the wording to ‘…showed typical retinal features of RP’

Line 210: suggest clarify the variant is in PRPH2 as you have just referred to 2 IMPG2 variants.

Line 221: It is difficult to argue that this is a different phenotype between eyes. Both are macular dystrophy. I would soften the statement to different severity.

Line 225: Suggest change ‘referred’ to ‘reported’

Line 240: Remove ‘related to severity.’

Was there a correlation between RP phenotype and type or location of variant?

Discussion:

Line 251: Spell out FJD abbreviation at first use.

Please include a statement of study limitations.

Methods:

Lines 346/7: Using ‘reason of referral’ to categorize patients may introduce significant inaccuracy depending on the referring source.  This should be included in your limitations.

Other comments:

All gene names should be in italicized font. The same goes for ‘in silico’ and other terminology in Latin.

I suggest you add to the discussion the importance of these genetic findings from a Spanish population considering the widespread Spanish diaspora internationally.
